# Lithographic SERS Aptasensor for Ultrasensitive Detection of SARS-CoV-2 in Biological Fluids

**DOI:** 10.3390/nano12213854

**Published:** 2022-11-01

**Authors:** Vladimir Kukushkin, Oganes Ambartsumyan, Anna Astrakhantseva, Vladimir Gushchin, Alexandra Nikonova, Anastasia Dorofeeva, Vitaly Zverev, Alexandra Gambaryan, Daria Tikhonova, Timofei Sovetnikov, Assel Akhmetova, Igor Yaminsky, Elena Zavyalova

**Affiliations:** 1Osipyan Institute of Solid State Physics of Russian Academy of Science, 142432 Chernogolovka, Russia; 2Department of Microbiology, Virology and Immunology, I.M. Sechenov First Moscow State Medical University, 125009 Moscow, Russia; 3Moscow Institute of Physics and Technology, 141701 Dolgoprudny, Russia; 4N. F. Gamaleya Federal Research Center for Epidemiology & Microbiology, 123098 Moscow, Russia; 5Mechnikov Research Institute of Vaccines and Sera, 105064 Moscow, Russia; 6Chumakov Federal Scientific Center for Research and Development of Immune and Biological Products RAS, 108819 Moscow, Russia; 7Chemistry Department, Lomonosov Moscow State University, 119991 Moscow, Russia; 8Faculty of Physics, Lomonosov Moscow State University, 119991 Moscow, Russia

**Keywords:** SARS-CoV-2, COVID-19, SERS, lithography, aptamer, biosensor

## Abstract

In this paper, we propose a technology for the rapid and sensitive detection of the whole viral particles of SARS-CoV-2 using double-labeled DNA aptamers as recognition elements together with the SERS method for detecting the optical response. We report on the development of a SERS-aptasensor based on a reproducible lithographic SERS substrate, featuring the combination of high speed, specificity, and ultrasensitive quantitative detection of SARS-CoV-2 virions. The sensor makes it possible to identify SARS-CoV-2 in very low concentrations (the limit of detection was 100 copies/mL), demonstrating a sensitivity level comparable to the existing diagnostic golden standard—the reverse transcription polymerase chain reaction.

## 1. Introduction

The global economic and health crises caused by COVID-19 stressed the important role of rapid, specific diagnostic tests in minimizing the number of disease contraction cases and deaths in the population and in limiting the spread of the pandemic. The typical symptoms of coronavirus infection include cough, fever, and shortness of breath, although it often leads to severe pneumonia with the development of acute respiratory distress syndrome, ARDS [1]. Furthermore, after recovering from the disease, many have been suffering serious consequences, such as neurological disorders. Currently, the most common method applied in detecting SARS-CoV-2 and its variants is the reverse transcription polymerase chain reaction (RT-PCR). Because of its high specificity and sensitivity, RT-PCR is regarded as the golden standard in the diagnosis of COVID-19, enabling the detection of viral RNA at an early stage of the disease. However, this method has some disadvantages [2]. As the quality of sample transportation, storage, and extraction of viral RNA can significantly affect the results of the test, all the procedures have to be carried out by experienced specialists [3]. What is more, performing RT-PCR requires designated areas in the laboratory, expensive reagents, and can take several hours [3,4]. Another conventional approach in diagnosing COVID-19 is to use serological assays, such as the enzyme-linked immunosorbent assay (ELISA), the chemiluminescence immunoassay (CLIA), and the lateral flow immunoassay (LFIA) [4,5].

All the aforementioned technologies rely on optical spectroscopy or colorimetry as a detection method, for example, photoluminescence spectroscopy, chemoluminescence spectroscopy or absorption spectroscopy. However, wide spectral bands registered by these techniques are not sufficiently specific and may overlap with extraneous optical signals, increasing the limit of detection.

In the present paper, we propose to use another type of optical spectroscopy—surface-enhanced Raman spectroscopy (SERS), which is more sensitive compared to luminescence or colorimetric methods. What makes SERS attractive is the absence of spectral overload in the Raman scattering signal from the test sample, enabling multiplex spectral analysis for the detection of several targets with a single SERS sensor [6,7].

There have been published a few SERS-based biosensors for SARS-CoV-2 detection.

For example, Chen et al. developed a diagnostic SERS platform for rapid detection of SARS-CoV-2 and influenza virus [8], where two types of labeled aptamers to SARS-CoV-2 and influenza are immobilized on the Au «nanopopcorn» SERS-substrate in order to selectively react to each virus approaching the surface. Thus, the variation in the concentration of viruses (SARS-CoV-2 and influenza) changes the intensity of SERS signals from the corresponding labeled aptamers due to their removal from the reinforcing surface.

Yang et al. [9] reported a SERS-biosensor based on the angiotensin-converting enzyme 2 (ACE2)-modified Au, “virus traps”, that quickly detects SARS-CoV-2 in polluted water.

Pramanik et al. [10] introduced anti-spike antibody-coupled AuNPs as SERS probes capable of detecting COVID-19 viral antigens within 5 min. In the presence of the SARS-CoV-2 antigen, the AuNPs form clusters, allowing antigen detection at concentrations as low as 1 ng/mL.

Zhang et al. [11] devised a rapid-detection SERS biosensor, where the SERS probe is prepared by incorporating calcium ions and acetonitrile into the AgNPs-reinforced substrate. The analysis was also performed on clinical specimens, saliva and serum. The limit of detection (LOD) was found to be 5 × 10^3^ copies/mL for human adenovirus 3 and SARS-CoV-2. In the case of clinical specimens, the analysis required approximately 60 min.

In our previous work, we also used a colloidal type of aptasensor for the detection of SARS-CoV-2 [12]. In particular, we proposed a SERS aptasensor based on a colloidal solution, providing a combination of rapidity and specificity in the quantitative determination of SARS-CoV-2. However, the sensitivity of the method was not very high, with the detection limit of 5.5 × 10^4^ TCID_50_/mL. In fact, none of the techniques described above achieved the RT PCR limit of detection of 10^2^–10^3^ copies/mL given the time of point-of-care diagnostics of 15–20 min.

In essence, aptamers allow us to achieve high specificity of the detection while the SERS technique makes possible the creation of rapid sensors. At the same time, the reproducibility and sensitivity of the method strongly depend on the sensor design. In the case of colloidal nanoparticles, the non-reproducible operation of the sensor may be attributed to a number of factors, namely, the non-reproducibility of the average size and dispersion of the nanoparticles during the chemical synthesis, the uncontrolled process of their aggregation, the presence of impurities in the colloidal solution, etc. 

In the present study, we examined a periodic reproducible SERS sensor, obtained by physical methods of laser lithography, plasma-chemical etching of the substrate, and vacuum thermal deposition of thin metal films. This time, we used a lithographic SERS-aptasensor with a new test setup. The DNA aptamer to SARS-CoV-2 is attached to the surface of the SERS substrate through a thiol modification (SH), whereas the second modification Cyanine-3 (Cy3) enables the detection of changes in the optical signals (SERS and SEL—surface enhanced luminescence) according to the analyte content.

## 2. Materials and Methods

### 2.1. Reagents

In our experiments, we used the following reagents: extra clean acetone and isopropyl alcohol (AO Reachem, Moscow, Russia); phosphate-buffered saline (PBS) tablets (Ecoservice, Saint Petersburg, Russia); fetal calf serum (FCS); the fetal bovine serum (HyClone, Logan, UT, USA); L-glutamine and penicillin/streptomycin (PanEco, Moscow, Russia); glutaric aldehyde (CAS Number: 111-30-8, AppliChem GmbH, Darmstadt, Germany).

We also used four metals for evaporation, Cr, Ag, Au (GIRMET Ltd., Moscow, Russia), and Al (RD Mathis Company, Signal Hill, CA, USA). In lithography processes, we utilized AZ5214E photoresist (MicroChemicals, Ulm, Germany) and AZ 726 MIF Developer (Merck, Darmstadt, Germany). To optimize the nanoperiodic structure, we analyzed the intensity of the SERS signal from the 4-ABT (4-Aminobenzenethiol, CAS Number: 1193-02-8, Sigma-Aldrich, St. Louis, MO, USA) test molecules.

### 2.2. Lithographic SERS-Substrate

In our investigation, we used the SERS substrates formed into dielectric pillars with a period of 2170 nm and a height of 230 nm, coated with a thick metal layer. The pillar has a square cross-section, with rounded corners, close to 1300 nm on the side. The structures are manufactured by the following procedure. First, a 4 mm × 4 mm active region (containing pillars) was produced on a thermally oxidized silicon substrate, with the oxide thickness of 1200 nm, using maskless laser lithography µMLA (Heidelberg Instruments Mikrotechnik GmbH, Heidelberg, Germany). Next, the anisotropic reactive-ion etching of SiO_2_ in the mixture of SF_6_ and Ar gases was performed employing the Oxford Plasmalab System 100 (Oxford Instruments, Bristol, UK). Finally, the thin-film deposition was carried out using the NANO 38 system (Kurt J. Lesker Company, Jefferson Hills, PA, USA).

All the involved processes were conducted in the following order:Si/SiO_2_ substrates were cleaned with acetone and isopropyl alcohol in an ultrasonic bath FB15047 (Fisher Scientific, Loughborough, UK).A single layer of AZ5214E photoresist was applied using a spin coater Delta 6RC (SUSS MicroTec SE, Garching, Germany) with a spin rate of 6000 rpm. After a 4 min prebaking on a hotplate (Wenesco, Addison, IL, USA) at 90 °C, 1.13 µm thickness of the resist film was achieved. Positive photolithography is accomplished by using optical focusing throughout the process. The development process was carried out in AZ 726 MIF Developer, with deionized water as a stop solution.Inside the chamber, aluminum was thermally evaporated onto the samples at the pressure of more than 8 × 10^−6^ Torr at the deposition rate of 1 A/s to obtain a 50 nm layer.The substrates with the metal on the top surface were immersed in acetone for the lift-off. As a result, a metal mask in the form of future pillars was created on the samples for the subsequent reactive-ion etching. After the etching process at the pressure of more than 2 × 10^−5^ Torr, the aluminum was rinsed off in AZ 726 MIF Developer.The substrate was washed with acetone and isopropyl alcohol in an ultrasonic bath to remove the residues of the developer.The layers of Cr, Ag, and Au, with respective thicknesses of 7, 40, and 15 nm, were successively evaporated onto the samples at the pressure of more than 10^−7^ Torr at the rate of 1 A/s. This final step was performed just before taking the measurements.

### 2.3. Viruses

In our study, the SARS-CoV-2 virus was provided by N. F. Gamaleya Federal Research Center for Epidemiology and Microbiology. In particular, we used the Dubrovka strain (GenBank ID: MW161041.1). The virus was grown on a Vero E6 (ATCC CRL-1586) cell line. The cells were cultured at 37 °C in a complete Dulbecco’s modified Eagle’s medium (DMEM) with 10% fetal bovine serum (FBS), L-glutamine (4 mM), and penicillin/streptomycin solution (100 IU/mL; 100 µg/mL). The virus titer was calculated as described by Ramakrishnan et al. [13]. It was found to be 1.2 × 10^6^ TCID50/mL, determined as TCID50 by the endpoint dilution assay. The RNA detection of SARS-CoV-2 was carried out by means of a real-time PCR device, Rotor Gene 6000 (Corbett Research, Australia), using a set of reagents “SARS-CoV-2/SARS-CoV” (DNA Technology, Moscow). All the experiments with the live SARS-CoV-2 were conducted according to the approved standard operating procedures of the NRCEM biosafety, level-3 facility. Viruses were inactivated by 0.05% (*v*/*v*) glutaric aldehyde.

The influenza viruses were provided by the Chumakov Federal Scientific Center for Research and Development of Immune and Biological Products of the Russian Academy of Sciences. Two particular strains were investigated, the influenza A virus (IvA) A/FPV/Rostock/34 R6p (256 HAU/mL in stock solution), and the influenza B virus (IvB) B/Victoria/2/1987 (2000 HAU/mL in stock solution). The virus stocks were propagated in an allantoic cavity of 10-day-old embryonated specific pathogen-free chicken eggs. The eggs were incubated at +37 °C, cooled to +4 °C over the 48 h post-infection period, and harvested 16 h later. The study scheme was approved by the Ethics Committee of the Chumakov Institute of Poliomyelitis and Viral Encephalitides, Moscow, Russia (Approval №4 from 2 December 2014). The viruses were deactivated by 0.05% (*v*/*v*) glutaric aldehyde, preserved by adding 0.03% (*w*/*v*) NaN_3_ and stored at +4 °C.

The HEp-2 and HeLa cells were cultured in complete RPMI and DMEM media, respectively. The A2 strain of the human respiratory syncytial virus (RSV) and the type-3 of the human adenoviruses (ADV 3) were cultured on HEp-2 and HeLa cells, accordingly, in the medium containing 2% FCS. In brief, the day before infecting the HEp-2 and HeLa cells, they were seeded in 25 cm^2^ flasks and left to grow to confluence overnight at +37 °C and 5% CO_2_. The next day, the medium was removed from the flask, and the monolayer was gently washed with 5 mL of the serum-free medium. The cells were then infected with RSV or ADV. The flasks were left at +37 °C for 2 h, being gently shaken. After that, 10 mL of the medium containing 2% FCS was added to each flask, and they were left for another 36–48 h at +37 °C and 5% CO_2_. For the RSV-infected culture, the cells were harvested with a cell scraper, detaching 50% of the cells and revealing the formation of syncytia. Then, the cells were transferred into a 50 mL falcon tube to be sonicated for 3 min at +4 °C. For the ADV-infected culture, the cells were also harvested using a cell scraper, detaching 50% of the cells, followed by two freeze–thaw cycles at −80 °C. For both the RSV and ADV-infected cultures, the cells were cleaned by centrifugation at 4000 RPM for 5 min at +4 °C. The supernatants containing the inactivated virus were then stored at +4 °C. In the cases of RSV A2 and ADV 3, we used the viral stocks at 6.3 × 10^5^ and 2.9 × 10^6^ TCID_50_/mL, respectively. 

### 2.4. Aptamers and Their Assembly

The DNA aptamer to receptor-binding domain S-protein of SARS-CoV-2 with thiol (SH) and Cyanine-3 (Cy3) modifications: SH-RBD-1C-Cy3, (SH-C6)-5′-d(CAGCACCGACCTTGTGCTTTGGGAGTGCTGGTCCAAGGGCGTTAATGGACA)-3′-(Cy3), was synthesized by Synthol (Moscow, Russia).

The assembly of the aptamer structure was conducted according to the following algorithm. First, the aptamers were prepared in 1 µM concentrations in the PBS buffer at pH 7.4, with 8 mM of Na_2_HPO_4_, 1.5 mM of KH_2_PO_4_, 140 mM of NaCl, and 3 mM of KCl. The solutions were heated to 95 °C, maintained at that temperature for 5 min, and then cooled to room temperature. Finally, the solutions were diluted with PBS to produce a 200 nM solution of the aptamer. All the solutions were made with ultrapure water provided by Millipore (Merck Millipore, Burlington, MA, USA).

The SERS substrates were incubated in 20 µL of 200 nM solution of the aptamer SH-RBD-1C-Cy3 for 15 min. The aptamer solution was removed using a piece of filter paper. The substrate was rinsed with 20 µL of ultrapure water and the drop was dried off with a piece of filter paper. The aptamer-modified substrates were prepared prior to the experiment. 

### 2.5. SERS Measurements

The aptamer-modified SERS substrates was incubated with the virus for 10 min. The solution was removed using a piece of filter paper. The substrate was rinsed with 20 µL of ultrapure water and placed into a tube with 500 µL of ultrapure water. The measurements were performed in the drop of ultrapure water escaping desiccation of the surface.

The samples with the virus were prepared as follows. The stock solution of the Wuhan variant of SARS-CoV-2 (10^10^ copies/mL by PCR and 1.2 × 10^6^ TCID_50_/mL) was diluted successively 10^2^, 10^3^, 10^4^, 10^5^, 10^6^, 10^7^, 10^8^, and 10^9^ times. The culture fluid from the non-infected cells diluted 10^3^ times with PBS was used as a virus dilution medium to minimize the effect of the off-target interactions. The control viruses—the influenza A virus (1.6 × 10^5^ HAU/mL), the influenza B virus (2 × 10^4^ HAU/mL), RSV A2 (6.3 × 10^5^ TCID_50_/mL), and ADV 3 (2.9 × 10^6^ TCID_50_/mL)—were diluted in the same way.

The SERS spectra were acquired with an Olympus BX51 optical scanning microscope (Olympus Corporation, Japan) based on the RamanLife RL532 spectrometer (TeraSense Group, Inc., San Jose, CA, USA) with 532 nm laser wavelength and 5 mW output power. The spectrometer provided the spectral resolution of 4–6 cm^−1^ over the spectral range of 160–4000 cm^−1^, at the laser spot of 10 µm in diameter. The map of the SERS signal distribution from Cy3, 2 mm × 2 mm in size, was recorded over the entire area of the sample employing a 10× objective with the step of 200 µm in the XY-scanning mode. As a result, we obtained the maps of the integral intensity distribution for the Raman line in the spectral window of 1588 ± 50 cm^−1^ and the luminescence in the spectral window of 1520 ± 15 cm^−1^ (578.8 ± 0.4 nm). Then, we calculated the average values of the data as well as the measurement errors associated with each value.

### 2.6. Study of the Surface Topology

The surface morphology of the SERS substrates was investigated by way of scanning electron microscopy, using the scanning electron microscope Supra 50VP (Zeiss, Germany). The electron optical GEMINI column provides excellent beam brightness with an ultrahigh resolution of 1 nm at the accelerating voltage of 20 kV. The substrate surface and profiles were scanned at the accelerating voltage of 10 kV, using the aperture size of 30 µm, at the work distance of 8 mm, and the chamber pressure of 9 × 10^−4^ Pa.

In addition, the surface was examined by means of a FemtoScan atomic force microscope (Advanced Technologies Center, Russia) used in the resonant mode. In that case, we utilized high-resolution sharp silicon cantilevers fpn11 (resonant frequency about 150 kHz, mechanical rigidity of 5.3 N/m) and HA_HR (450 kHz and 34 N/m, respectively). Line frequency was installed at the level of 0.5 Hz. The signal processing and imaging were carried out using the FemtoScan Online software [14].

## 3. Results

### 3.1. Aptamer as a Reporter Molecule

As a recognition element in the biosensor, we made use of the aptamer RBD-1C for the receptor-binding domain of the S-protein in SARS-CoV-2 [15]. Previously, this aptamer was shown to bind not only the receptor-binding domain of the S-protein but also the whole SARS-CoV-2 virus [12]. It has a dumbbell structure (Figure 1) with the 5′- and 3′-ends of the aptamer pulled together at the center. According to molecular docking, the central part interacts directly with the receptor-binding domain of the S-protein [15]. In our study, we introduce two modifications into this region, namely, the thiol-group to the 5′-end and Cyanine-3 to the 3′-end, as indicated in Figure 1. 

### 3.2. Optimization of the SERS-Substrate

We have previously shown that the resonant amplification of Raman signals in nanoperiodic structures occurs at the metal–air interface provided that *p* ≈ λ × N, where *p* is the period of the structure, λ is the wavelength of the exciting radiation (532 nm), and N is a natural number [16]. Therefore, to ensure the transition to the mass production of inexpensive nanoperiodic structures, we employed laser lithography as it makes possible the reproduction of structures with a period *p* ≈ 4 × λ ≈ 2170 nm at the minimum possible gaps of d ≈ 550 nm.

As the next step, we optimized the nanostub height by increasing the h parameter from 50 to 1200 nm. Figure 2 shows the h-dependence of the integral intensity of the Raman line 1140 cm^−1^ in the spectral window of 1120–1170 cm^−1^ measured for the test substance 4-ABT (5 mW output laser power, the laser spot of 10 µm in diameter, 10× objective, exposure time 1 s).

The data make it evident that at least two geometric resonances associated with the formation of standing plasmon-polariton waves are realized in these structures. The standard condition for the formation of standing waves imposes a certain requirement on the h-parameter, namely: h = λ · (*n* + 1/2). Thus, the first geometric resonance (*n* = 0) is observed at h ≈ 250 nm and the second (*n* = 1) at h ≈ 800 nm. For our experiments, we chose a height of about 250 nm (to be exact, structures with a column height of 230 nm were obtained). At the chosen height, the columns remain intact during the ultrasonic processing, making possible the repeated use of the substrate in creating sensors.

At the final optimization stage, we considered the deposition of metal layers onto the surface of the developed nanoperiodic substrate. In order for the surface to be resistant to the effects of the protein biological assays and to withstand all the stages of adsorption, a 7 nm adhesive layer of chromium was deposited as the primary metallic film, followed by the successive deposition of silver and gold, at the respective layer thicknesses of 40 and 15 nm. This particular combination of metal layers provides the optimal optical properties of the substrate at 532 nm wavelength of the excitation laser radiation. It also allows for the substrate to be modified by the aptamers with thiol groups.

The Raman topography of 4-ABT (5 mW output laser power, the laser spot of 10 µm in diameter, 10× objective, exposure time 1 s) of the optimized substrate was determined almost over the entire SERS-active zone, yielding a map of 5 × 5 points with 600-micron increments. The results of this experiment demonstrate the uniformity of the enhancement distribution of the Raman signal over the surface of the structure (Figure 3). Based on the data, we found that the maximum deviation from the average intensity of the SERS signal did not exceed 10%. This is a good indicator, given that not only the surface of the substrate can contribute to the error in the SERS signal but also the fluctuations in the laser power level as well as the heterogeneity of the distribution of the test substance over the surface.

The enhancement factor was measured by comparing the SERS intensity from the test substance (4-ABT) on the developed substrate and on a substrate (Blue & Green Substrate, Enhanced Spectrometry, Inc., Meridian, ID, USA) with a known enhancement factor of 2.2 × 10⁶. The SERS signal on the developed optimized substrate (with the same signal registration parameters) was 7.4 times larger than on a substrate with a known enhancement factor. Thus, the enhancement factor was: K = 1.6 × 10^7^.

### 3.3. Lithographic SERS-Aptasensor

The conditions required for SH-RBD-1C-Cy3 aptamer immobilization were optimized by analyzing the ionic strength, time, and temperature regimens. We observed that the SERS and SEL spectra intensities were 10 times higher for the aptamer immobilization conducted from phosphate-buffered saline (PBS) compared to the aptamer sample in water. We also found that gradual heating from 20 °C to 37 °C slightly increased the signal. Hence, as an optimal combination, we chose incubation with PBS solution of the aptamer at room temperature. Furthermore, our experimental results evidence that SERS and SEL intensities are greatly affected by incubation time and aptamer concentration (Figure 4). Hence, considering the SERS/SEL ratio as an analytical signal that depends on the quantity of immobilized aptamer and the distance between the label and the metal surface, we chose 200 nM of aptamer and quick immobilization as the optimal conditions.

We have tested the sensor assembly scheme involving a double-labeled aptamer based on the nanoperiodic solid-state substrates developed for detecting SARS-CoV-2. Previously, we have shown that modification of either 5′- or 3′-end retains the affinity of the aptamer to the receptor-binding domain of the S-protein in the nanomolar range (K_D_ were 6 nM and 27 nM, correspondingly) [17]. In the sensor, the aptamer SH-RBD-1C-Cy3 is anchored to the metal surface through the thiol group while the reporter (Cyanine-3) is fixed near the surface, producing a high Raman signal and a low fluorescence signal since the luminescence is extinguished near the metal [18,19]). Figure 5 shows the concentration dependency of the SERS to SEL signal ratio for the studied SARS-CoV-2 virus in comparison to the control viruses (influenza A and influenza B).

The data presented in the figure indicate that with an increasing concentration of SARS-CoV-2, the SERS/SEL index shows a gradual decrease from 0.47 to 0.27. As the SARS-CoV-2 concentration is reduced, the standard deviation of the studied index increases due to the greater heterogeneity of the signal distribution map over the surface. As for the control viruses, no noticeable dependence on concentration is found in the behavior of the corresponding index while its values exceeded the 0.43 level regardless of the concentration of viruses (Figure 5, Table 1).

When the virus interacts with the aptamer, the reporter is oriented into the complex with S-protein leading to the decrease of the intensity of the reporter’s Raman signal and an increase in the luminescence signal. According to the measurement results, the given sensor enables the detection of SARS-CoV-2 with a sensitivity of 100 copies/mL, which is comparable to the limit of detection for the polymerase chain reaction, LOD = 10^2^–10^3^ copies/mL [20,21]. Thus, the experimental evidence demonstrates that the proposed sensor is superior in sensitivity to the rapid antibody-based assays (LOD = 10^6^ copies/mL) [22].

Figure 6 and Figure 7 demonstrate that SARS-CoV-2 virions are deposited on the surface of the developed nanoperiodic SERS sensors, whereas the control viruses are not, due to the high specificity of the SH-RBD-1C-Cy3 aptamer.

## 4. Discussion

Recently, aptamer-based analysis methods have been actively developed due to their low cost, high stability, ease of modification, and superior specificity and affinity. Combining the surface enhancement of Raman scattering and luminescence with molecular recognition through aptamers has a promising future as it allows to create effective and inexpensive sensor designs for detecting molecular targets of various nature, such as low-molecular-weight compounds [23], protein toxins [24], bacteria [25], viruses [12,26], and circulating cancer cells [27,28].

Despite the high sensitivity and specificity of the existing diagnostic aptamer SERS platforms, their non-reproducible operation is a serious obstacle to their mass distribution. Substrates with chaotic topography based on nanostructures or colloidal solutions of nanoparticles are not reproducible and unstable during experiments. The colloidal particles exhibit uncontrolled size variation during the synthesis and strongly depend on the reagent purity and the synthesis recipe. Moreover, they can randomly form aggregates in the buffer media. On the other hand, the nanostructured substrates do not have a microscopically homogeneous distribution of the Raman signal amplification over the surface. During sorption experiments with biological media, they are subject to changes in the surface morphology due to the ‘mobility’ of the silver nanoparticles [29]. As a result of the modified nanoparticle morphology, the plasmon resonance of the SERS structures shifts, altering the optimal conditions for substrate excitation.

In the present paper, we reported on a nanoperiodic SERS substrate with a homogeneous enhancement of the Raman scattering over the surface. Following the step-by-step optimization of the structure geometry, we developed a low-cost, reproducible and reusable SERS substrate that works effectively at 532 nm laser excitation. The aptamer sensor based on this structure makes it possible to detect SARS-CoV-2 from the group of respiratory viral infections with a LOD = 100 copies/mL within 12 min.

Further cost reduction and miniaturization of Raman systems will make feasible their use as point-of-care devices and make them widely applicable in medical diagnostic practice in combination with aptamer sensors.

## Figures and Tables

**Figure 1 nanomaterials-12-03854-f001:**
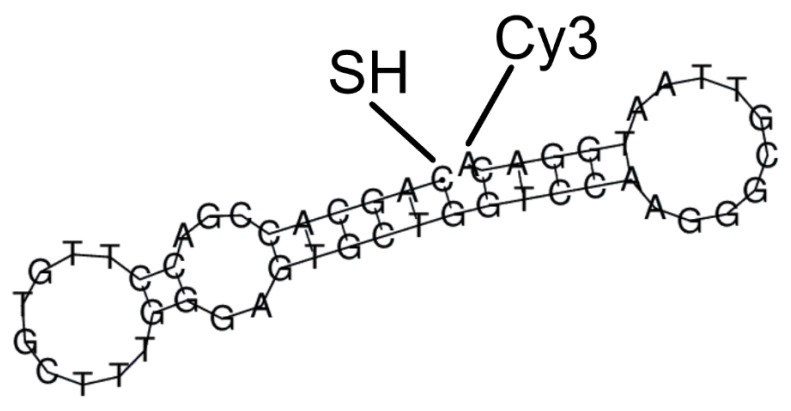
The SH-RBD-1C-Cy3 aptamer structure.

**Figure 2 nanomaterials-12-03854-f002:**
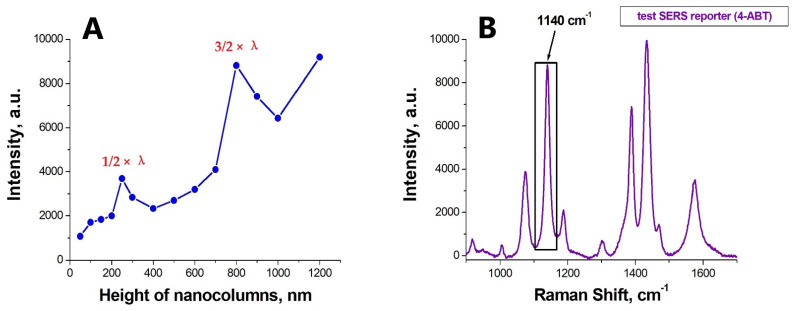
The effect of the dielectric nanocolumn height on the Raman signal amplification. (**A**) the dependency of the integral intensity for the 4-ABT substance line in the spectral window of 1120–1170 cm^−1^, measured at 80 nm thickness of the deposited silver layer on top of the nanocolumns. (**B**) The SERS spectrum for 4-ABT, obtained from a nanoperiodic structure, with the column height of 250 nm and the thickness of the deposited silver of 80 nm.

**Figure 3 nanomaterials-12-03854-f003:**
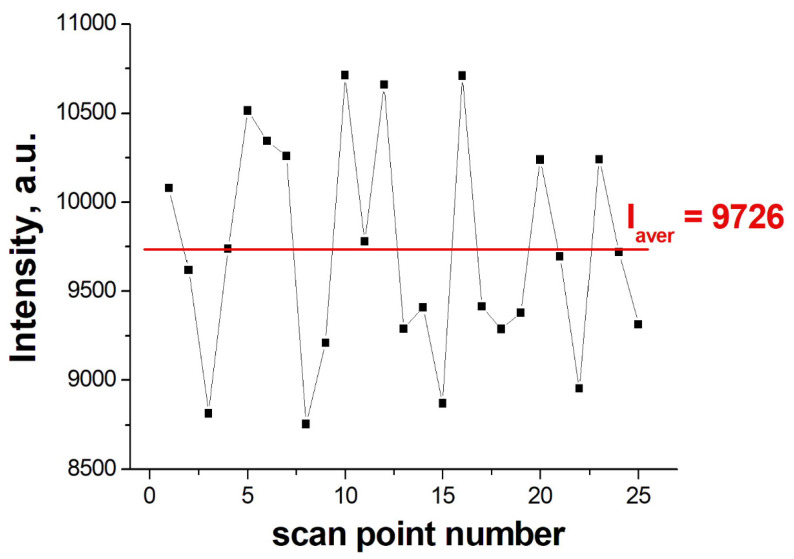
Distribution of the SERS signal over the substrate surface. The relative standard deviation was 6.2%.

**Figure 4 nanomaterials-12-03854-f004:**
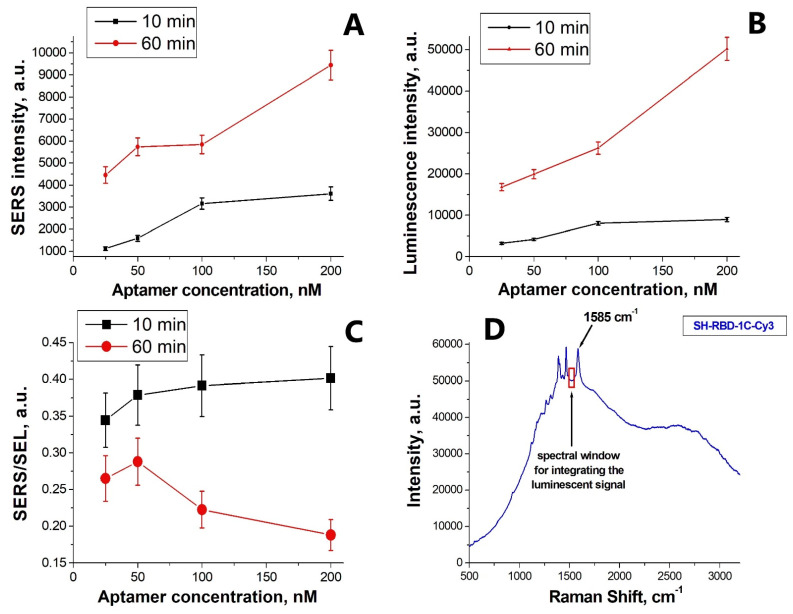
The effect of the concentration of the SH-RBD-1C-Cy3 aptamer on the SERS and SEL optical signals from the Cy3 label, observed for the incubation times of 10 and 60 min. (**A**) Variation in the SERS signal. (**B**) Variation in luminescence (expressed as a volume luminescence signal and a surface-enhanced luminescence signal). (**C**) Variation in the ratio of the SERS signal to the luminescence signal. (**D**) SERS spectrum of the SH-RBD-1C-Cy3 aptamer.

**Figure 5 nanomaterials-12-03854-f005:**
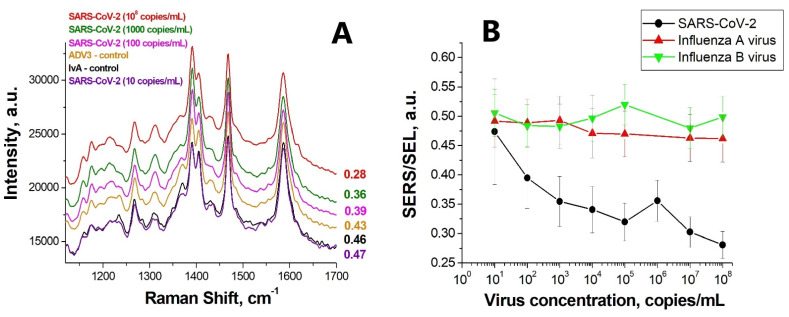
(**A**) Surface-enhanced Raman scattering and luminescence spectra for control viruses and SARS-CoV-2 at different concentrations. (**B**) Dependence of the SERS/SEL parameter on the concentrations of SARS-CoV-2, the influenza A virus, and the influenza B virus.

**Figure 6 nanomaterials-12-03854-f006:**
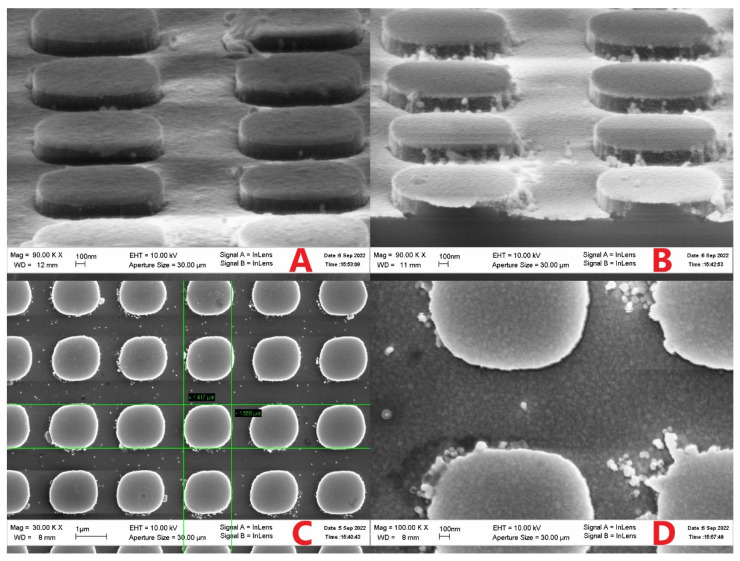
SEM images of the lithographic SERS structures with the aptamer SH-RBD-1C-Cy3. (**A**) The structure with the control influenza A virus. (**B**–**D**) The structures with SARS-CoV-2.

**Figure 7 nanomaterials-12-03854-f007:**
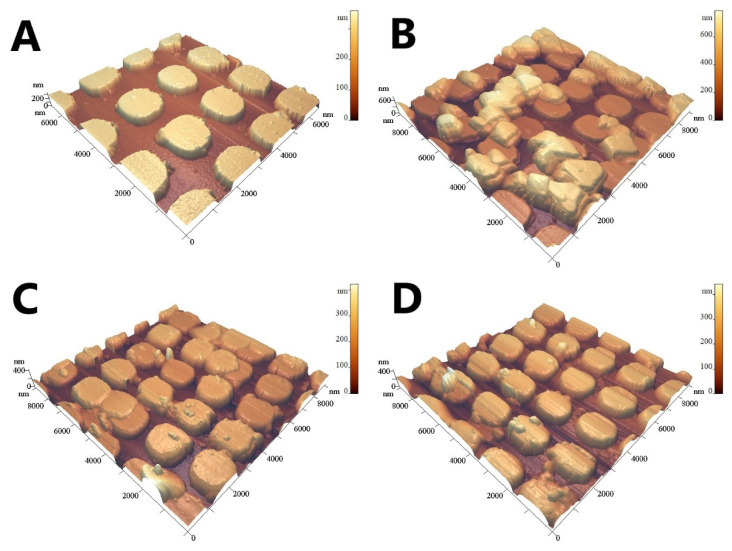
AFM images of the lithographic SERS structures with the aptamer SH-RBD-1C-Cy3. (**A**) The structure with the control influenza A virus. (**B**–**D**) The structures with SARS-CoV-2.

**Table 1 nanomaterials-12-03854-t001:** SERS/SEL parameter for control viruses (ADV 3 and RSV 2) and SARS-CoV-2 at different concentrations.

Type 3 of Human Adenoviruses (ADV 3)		Human Respiratory Syncytial Virus A2 Strain (RSV A2)		SARS-CoV-2	
Concentration, **TCID_50_/mL**	SERS/SEL, a.u.	Concentration, **TCID_50_/mL**	SERS/SEL, a.u.	Concentration, **TCID_50_/mL**	SERS/SEL, a.u.
2.9	0.43 ± 0.03	0.63	0.51 ± 0.05	1.2	0.34 ± 0.04
29	0.47 ± 0.04	6.3	0.44 ± 0.04	12	0.32 ± 0.03
290	0.52 ± 0.04	63	0.47 ± 0.04	120	0.36 ± 0.03
2900	0.43 ± 0.03	630	0.48 ± 0.04	1200	0.30 ± 0.03

## Data Availability

Not applicable.

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
