# Peer review of "Lithographic SERS Aptasensor for Ultrasensitive Detection of SARS-CoV-2 in Biological Fluids"

_nanomaterials, 2022, doi:10.3390/nano12213854_

Round 1
Reviewer 1 Report
The manuscript describes a new SERS aptasensor for SARS-CoV-2. This is a method that is in urgent need considering the impact of COVID-19. The manuscript can be published in Nanomaterials after minor revisions as noted below. 1. How do the authors differentiate between different types of viruses using SERS signals (Figure 5)? 2. The authors might better present the standard deviations (error bars) for the data presented in Figure 4. 3. Have the authors tested the binding affinity of aptamer they used on this specific detection platform?
Author Response
Thanks for your comments!
Q1. How do the authors differentiate between different types of viruses using SERS signals (Figure 5)?
A1. The position of the Raman lines in the SERS spectra in the developed sensor is the same for all types of viruses under study, since the signal comes from the Cy3 label on the aptamer to SARS-CoV-2 fixed on the surface of the lithographic SERS substrate. When detecting SARS-CoV-2, the aptamer interacts with the virus, the Cy3 is oriented into the complex with S-protein leading to the decrease of the intensity of reporter`s Raman signal and an increase in the luminescence signal. The SERS/SEL index gradually decreases from 0.47 to 0.27.
As for the control viruses, no noticeable dependence on concentration is found in the behavior of the corresponding index while its values exceeded the 0.43 level regardless of the concentration of viruses.
In the new version of the article, we added the Figure 5B with SERS spectra for different viruses. It shows how the SERS/SEL index changes.
Q2. The authors might better present the standard deviations (error bars) for the data presented in Figure 4.
A2. We have added error bars for the data presented in Figure 4.
Q3. Have the authors tested the binding affinity of aptamer they used on this specific detection platform?
A3. Previously, we have shown that modification of either 5’- or 3’-end retains the affinity of the aptamer to the receptor-binding domain of the S-protein in the nanomolar range (KD were 6 nM and 27 nM, correspondingly) [DOI:10.3390/ijms23010557].
The discussion of this question has been added to the text of the article.
Reviewer 2 Report
Recommendation – Minor Revision
Kukushkin et al., developed a SERS aptasensor for the detection of whole viral particles of SARS-CoV-2 using double-labeled DNA aptamers as recognition elements. The manuscript can be accepted after a minor revision.
1) In the reagent section (2.1), the authors can provide more details about the chemicals and reagents with company details.
2) The authors can include the measurement conditions for the 4-ABT molecule.
3) The authors mentioned that PBS buffer and higher temperature (37 0C) enhanced the SERS signals. Why is it so?
4) The authors can provide the SERS spectra of the aptamer for different concentrations of SARS-CoV-2, the influenza A virus, and the influenza B virus.
5) The enhancement factor of the SERS substrate can be included in the manuscript.
Author Response
Thanks for your comments!
Q1. In the reagent section (2.1), the authors can provide more details about the chemicals and reagents with company details.
A1. We have added information about reagents in the text of the article in paragraph 2.1.
«In our experiments, we use the following reagents: extra clean acetone and isopropyl alcohol (AO Reachem, Russia); phosphate-buffered saline (PBS) tablets (Ecoservice, Russia); the fetal calf serum (FCS), the fetal bovine serum (HyClone, UT, USA); L-glutamine and penicillin/streptomycin (PanEco, Russia); glutaric aldehyde (CAS Number: 111-30-8, AppliChem GmbH, Germany)»
PBS was in tablet form and included 8 mM of Na2HPO4, 1.5 mM of KH2PO4, 140 mM of NaCl, and 3 mM of KCl.
Q2. The authors can include the measurement conditions for the 4-ABT molecule.
A2. We have added information in the text of the article in paragraph 3.2.
«Fig. 2 shows the h-dependence of the integral intensity of the Raman line 1140 cm-1 in the spectral window of 1120-1170 cm-1 measured for the test substance 4-ABT (5 mW output laser power, the laser spot of 10 µm in diameter, 10x objective, exposure time 1s)».
«The Raman topography of 4-ABT (5 mW output laser power, the laser spot of 10 µm in diameter, 10x objective, exposure time 1s) of the optimized substrate was determined almost over the entire SERS-active zone, yielding a map of 5 x 5 points with 600-micron increments».
Q3. The authors mentioned that PBS buffer and higher temperature (370C) enhanced the SERS signals. Why is it so?
A3. In our opinion, the probable reason for the increase in the signal is an increase in the diffusion rate of the aptamer to the nanostructured surface.
Q4. The authors can provide the SERS spectra of the aptamer for different concentrations of SARS-CoV-2, the influenza A virus, and the influenza B virus.
A4. In the text of the article, we added Figure 5A with SERS spectra for different targets (for control viruses and SARS-CoV-2). It is impossible to get all the spectra in one picture, because they are essentially the same Cy3 spectrum with a changing index of the ratio of SERS to SEL. The change in this index occurs when the aptamer interacts with SARS-CoV-2 and this index does not change in the presence of control viruses with which the aptamer does not interact.
Q5. The enhancement factor of the SERS substrate can be included in the manuscript.
A5. We have added information in the text of the article in paragraph 3.2.
«The enhancement factor was measured by comparing the SERS intensity from the test substance (4-ABT) on the developed substrate and on a substrate (Blue & Green Substrate, Enhanced Spectrometry, Inc., ID, USA) with a known enhancement factor 2.2 * 10⁶. The SERS signal on the developed optimized substrate (with the same signal registration parameters) was 7.4 times larger than on a substrate with a known enhancement factor. Thus, the enhancement factor was: K = 1.6 * 107».
Reviewer 3 Report
The paper is competely missing raw experimental data (i.e. Raman spectra obtained on different samples) without leaving the possibility to critically judge the quality of the obtained signals and so on. I think the manuscript it should be improved in this sense before publication.
Author Response
Thanks for your comments!
In the text of the article, we added Figure 5A with SERS spectra for different targets (for control viruses and SARS-CoV-2). It is impossible to get all the spectra in one picture, because they are essentially the same Cy3 spectrum with a changing index of the ratio of SERS to SEL.
The change in this index occurs when the aptamer interacts with SARS-CoV-2 and this index does not change in the presence of control viruses with which the aptamer does not interact.
In the sensor, the aptamer SH-RBD-1C-Cy3 is anchored to the metal surface through the thiol group while the reporter (Cyanine-3) is fixed near the surface, producing a high Raman signal and a low fluorescence signal since the luminescence is extinguished near the metal.
Round 2
Reviewer 3 Report
Paper is now ready for publication.